# Cheungsam Seed Husk Extract Reduces Skin Inflammation through Regulation of Inflammatory Mediator in TNF-α/IFN-γ-Induced HaCaT Cells

**DOI:** 10.3390/plants13121704

**Published:** 2024-06-19

**Authors:** Ji-Ye Han, Yun Jung Lee, Do-Won Lim, Hyun-Ju Jung, EunJeong Kwon, Jongki Hong, Young-Mi Lee

**Affiliations:** 1Department of Oriental Pharmacy, College of Pharmacy and Wonkwang-Oriental Medicines Research Institute, Wonkwang University, Iksan 54538, Republic of Korea; hsue0112@gmail.com (J.-Y.H.); shrons@wku.ac.kr (Y.J.L.); limsc05@wku.ac.kr (D.-W.L.); hyun104@wku.ac.kr (H.-J.J.); 2College of Pharmacy, Kyung Hee University, Seoul 02447, Republic of Korea; dmswjd4876@naver.com (E.K.); jhong@khu.ac.kr (J.H.)

**Keywords:** *Cannabis sativa* L., Cheungsam, skin inflammation, atopic dermatitis

## Abstract

Cannabis contains numerous natural components and has several effects such as anticancer, anti-inflammatory and antioxidant. Cheungsam is a variety of non-drug-type hemp, developed in Korea and is used for fiber (stem) and oil (seed). The efficacy of Cheungsam on skin is not yet known, and although there are previous studies on Cheungsam seed oil, there are no studies on Cheungsam seed husk. In this study, we investigated the potential of Cheungsam seed husk ethanol extract (CSSH) to alleviate skin inflammation through evaluating the gene and protein expression levels of inflammatory mediators. The results showed that CSSH reduced pro-inflammatory cytokines (IL-1β, IL-6, IL-8, MCP-1 and CXCL10) and atopic dermatitis-related cytokines (IL-4, CCL17, MDC and RANTES) in TNF-α/IFN-γ-induced HaCaT cells. Furthermore, ERK, JNK and p38 phosphorylation were decreased and p-p65, p-IκBα, NLRP3, caspase-1, p-JAK1 and p-STAT6 were suppressed after CSSH treatment. CSSH significantly increased the level of the skin barrier factors filaggrin and involucrin. These results suggest that Cheungsam seed husk ethanol extract regulates the mechanism of skin inflammation and can be used as a new treatment for skin inflammatory diseases.

## 1. Introduction

*Cannabis sativa* L. is a native plant to Central Asia and contains hundreds of natural products such as cannabinoids, terpenoids, steroids, flavonoids and alkaloids [1]. These natural products have antinociceptive, anxiolytic, antipsychotic, anti-inflammatory and antioxidant effects and are used to treat epilepsy and multiple sclerosis [2,3,4]. Cannabinoids are divided into phytocannabinoids and endocannabinoids, with phytocannabinoids including cannabidiol (CBD), Δ9-tetrahydrocannabinol (THC), cannabigerol (CBG) and cannabinol (CBN), among others; the effects of CBD and THC are best known [5,6,7]. Particularly, CBD has beneficial effects on various diseases such as Parkinson’s disease, Huntington’s disease, Dravet syndrome and Lennox–Gastaut syndrome [8]. Despite the effects of cannabinoids, these are legally used for medical purposes only in some countries because of the psychoactive effect of THC [9]. For this reason, new varieties of cannabis with low THC contents, such as Futura 75, Cherry Blossom, and Stormy D, have been developed [10]. Cheungsam is a new variety of hemp developed in Korea as a non-drug-type (low THC content) that is a hybrid IH3 and local Korean variety [11]. The pharmacologically active compounds of *C. sativa*, such as CBD and THC, are abundant in leaves and inflorescences; however, only non-addictive parts of *C. sativa* can be used for industrial or food purposes in Korea. Cheungsam stems are used as fibers and seeds as oils [12].

Inflammation is an innate immune response against infection and tissue damage and is induced by several cytokines such as TNF-α, IL-1β, IL-6 and MCP-1 [13]. Inflammation is mainly mediated by polymorphonuclear leukocytes and the activation of the inflammasome, MAPKs, NF-κB and JAK/STAT signaling pathways [14]. The chronicization of inflammation results in tissue injury and fibrosis, and typical chronic inflammatory diseases include inflammatory bowel disease, arthritis, diabetes, Alzheimer’s disease and atopic dermatitis (AD) [15].

Atopic dermatitis is a highly recurrent and incurable chronic inflammatory skin disease. This disease mostly occurs in children and is accompanied by itching and skin lichenification. Immunologically, AD is a hypersensitivity reaction and is associated with T cells. Cytokines/chemokines released from immune cells affect the development of AD [16]. AD is characterized by a Th1/Th2 imbalance and Th2-dominant immune response during the acute phase [17]. The activation of Th2 cells induces the secretion of Th2 cytokines (IL-4, IL-5, IL-10, IL-13 and IL-31). Secreted IL-4 and IL-13 are key biomarkers in AD and mediate IgE production, Th2 cell differentiation and skin barrier protein expression [18,19,20]. Additionally, several chemokines are associated with AD, and the upregulation of CCL17, MDC and RANTES are representative biomarkers of this disease [21].

Skin barrier dysfunction is another characteristic of AD. The skin acts as a physical barrier against infection and damage from the outside environment and is involved in body temperature control and sensory reception. The skin consists of the epidermis, dermis and subcutaneous layer, and the epidermis is composed of the stratum corneum, stratum granulosum, stratum spinosum and stratum basale. Furthermore, filaggrin, involucrin, and loricrin, which form a skin barrier, are located in the epidermis of keratinocytes and melanocytes [22,23,24]. Functional abnormalities of these proteins (such as mutation, expression reduction, and incorrect precursor processing) are commonly observed in AD tissues and are important features of AD [25,26].

Studies on improving skin inflammation using seeds and seed husk of medicinal plants, including *Swietenia macrophylla*, *Nelumbo nucifera* and *Cornus officinalis*, have been reported in previous studies [27,28,29]. Additionally, antioxidant and anti-inflammatory effects of *C. sativa* have been reported in human keratinocytes [30,31]. Despite many studies on *C. sativa*, most studies focused on the leaves and flowers of cannabis, with no studies on the seed husk [32,33,34]. Moreover, the mechanism of action underlying skin inflammatory disease remains unclear. In this study, we explored the effects of Cheungsam seed husk against atopic dermatitis and investigated how it affects skin inflammation.

## 2. Results

### 2.1. Gas Chromatography-Mass Spectrometry Analysis of Cheungsam Seed Husk Ethanol Extract

The final yield of Cheungsam seed husk ethanol extract (CSSH) after reflux extraction was 68.3%, and GC-MS was performed to analyze indicator components. The peaks of CBD, CBDA, Δ9-THC and CBN were detected in CSSH, and the cannabinoid content of CSSH per weight of dry powder was shown in Table 1 (Figure 1).

### 2.2. CSSH Decreases Pro-Inflammatory Cytokines and Chemokines in TNF-α/IFN-γ-Induced HaCaT Cells

To confirm the cytotoxicity of CSSH in HaCaT cells, we treated HaCaT cells with 1, 2, 5, 10, 20, 50, 100, 200 and 500 µg/mL of CSSH for 24 h and performed cell viability assay. As a result, CSSH showed no cytotoxicity at below 200 μg/mL (Figure 2A). To evaluate the anti-inflammation effects of CSSH, we investigated the mRNA levels of pro-inflammatory cytokines and chemokines using real-time PCR analysis. The mRNA levels of pro-inflammatory cytokines and chemokines (IL-1β, IL-6, IL-8, MCP-1 and CXCL10) were increased in TNF-α/IFN-γ-induced HaCaT cells but significantly decreased in a dose-dependent manner in HaCaT cells pre-treated with CSSH (25, 50, 100 and 200 ng/mL) (Figure 2B–F).

### 2.3. CSSH Reduces Atopic Dermatitis-Related Cytokines and Chemokines in TNF-α/IFN-γ-Induced HaCaT Cells

To determine the mechanism by which CSSH affects AD, we evaluated the mRNA levels of AD-related cytokines and chemokines in HaCaT cells. The cells were pre-treated with 25, 50, 100 and 200 ng/mL CSSH, and then inflammation was induced through TNF-α and IFN-γ stimulation. The mRNA expression levels of IL-4, CCL17, MDC and RANTES were reduced in a dose-dependent manner by CSSH, with the mRNA expression of RANTES showing the largest decrease (Figure 3).

### 2.4. CSSH Downregulates the MAPK (ERK, JNK and p38) and NF-κB Pathways in TNF-α/IFN-γ-Induced HaCaT Cells

We investigated whether CSSH regulates the MAPK and NF-κB pathways in TNF-α/IFN-γ-induced HaCaT cells. The phosphorylation of MAPKs was decreased at all concentrations following CSSH treatment. Particularly, the phosphorylation of JNK was reduced by 88% in the high-concentration CSSH (200 ng/mL)-treated group compared with that in the TNF-α/IFN-γ-treated group (Figure 4A,C). In addition, p65 and p-p65 in the nucleus were reduced in the CSSH-treated group, as were p-IκBα and IκBα in cytoplasm (Figure 4B,D).

### 2.5. CSSH Inhibits NLRP3 Inflammasome Activation in TNF-α/IFN-γ-Induced HaCaT Cells

To research NLRP3 inflammasome inhibitory effect of CSSH, we evaluated the protein levels of NLRP3 and caspase-1 in TNF-α/IFN-γ-induced HaCaT cells using Western blot analysis. The protein levels of NLRP3 and caspase-1 were greatly reduced following treatment with 100 and 200 ng/mL CSSH in TNF-α/IFN-γ-induced HaCaT cells (Figure 5).

### 2.6. CSSH Suppresses Phosphorylation of JAK1 and STAT6 in TNF-α/IFN-γ-Induced HaCaT Cells

To confirm the change in the JAK1/STAT6 pathway induced by CSSH, we performed Western blotting. JAK1 and STAT6 phosphorylation decreased in the CSSH-treated group. Furthermore, 100 and 200 ng/mL CSSH significantly reduced the phosphorylated form/total form ratio of JAK1 and STAT6 (Figure 6).

### 2.7. CSSH Upregulates Filaggrin and Involucrin in TNF-α/IFN-γ-Induced HaCaT Cells

We explored whether CSSH protects the skin barrier via the upregulation of filaggrin and involucrin in TNF-α/IFN-γ-induced HaCaT cells. CSSH inhibited the downregulation of filaggrin at all concentrations and increased involucrin at 50, 100 and 200 ng/mL (Figure 7).

## 3. Discussion

AD is a skin disease in childhood that is difficult to treat and is accompanied by other allergic diseases such as asthma and allergic rhinitis [35]. Several drugs have been developed for the prevention and treatment of AD; however, these drugs are mostly topical corticosteroids that have serious adverse effects such as skin thinning, osteoporosis, adrenal insufficiency and hypertension [36,37]. Therefore, treatments with fewer side effects are needed; natural product-derived compounds are good therapeutic candidates. In this study, we explored the anti-inflammatory effects of Cheungsam on human keratinocytes using Cheungsam seed husk extract (CSSH) and wanted to find out the role of CSSH on skin inflammatory diseases and the availability of CSSH as an active material.

*C. sativa* has been used as a folk remedy for a long time and has effects of anticancer, anti-inflammation, antioxidant, antinociceptive and anti-insomnia [38]. In previous studies, *C. sativa* extract significantly reduced the levels of intracellular ROS and inhibited pro-inflammatory cytokines/chemokines expression and NF-κB activation in HaCaT cells [30,31]. In particular, CBD, a major cannabinoid, improved psoriasis and alleviated skin damage, and the topical administration of CBD ointment improved the patient’s atopic dermatitis [39,40,41].

Many studies on *C. sativa* have been reported; however, most of them are studies on leaves and inflorescences [8,42,43,44]. Additionally, there are studies on hemp seed oil, but there are no studies on seed husk [45,46]. Furthermore, the effects of Cheungsam remain unclear. Cheungsam seed husk occurs naturally in the process of making Cheungsam seed oil and is disposed of as waste. Like the Cheungsam seed husk, the seed husk of other plants is also discarded as waste, but it has recently been reported that they have active compounds [47,48]. We initially investigated the cannabinoids present in CSSH and analyzed their contents. CBD, CBDA, THC and CBN were detected in CSSH. We also found that CBD is the major cannabinoid in CSSH because CBDA is decarboxylated into CBD under certain conditions (Figure 1 and Table 1).

Keratinocytes are the major cell type in the epidermis, and keratinocyte dysfunction is associated with chronic skin diseases. Keratinocytes regulate the immune response by producing immunosuppressive factors and cytokines [49,50]. In patients with AD, keratinocyte-derived pro-inflammatory cytokines such as IL-1β, IL-6, IL-8 and MCP-1 are increased, and AD-related chemokines including CCL17, MDC and RANTES are upregulated. These cytokines promote the infiltration of immune cells into lesions and induce skin barrier disruption [51]. RANTES/CCL5 plays an important role in allergic inflammation and is detected at high levels in the plasma of patients with AD. Likewise, the expression of TARC/CCL17 and MDC/CCL22 is significantly increased [52,53,54]. Thus, the downregulation of pro-inflammatory cytokines, RNATES/CCL5, TARC/CCL17 and MDC/CCL17 is an important requirement of AD treatment. Our data show that CSSH significantly reduced the levels of AD-related cytokines and chemokines (Figure 2 and Figure 3).

Several skin diseases, including AD, psoriasis, and seborrheic dermatitis, are associated with inflammation. MAPKs (ERK, JNK and p38) and NF-κB signaling are the main inflammation pathways that induce the production of pro-inflammatory cytokines [55,56]. Based on these mechanisms, we hypothesized that CSSH can diminish the phosphorylation of MAPKs and the translocation of NF-κB to the nucleus, and we confirmed the anti-inflammatory effect of CSSH in HaCaT cells (Figure 4). We further investigated how CSSH reduces AD-related cytokines and chemokines. As shown in Figure 2, the level of IL-1β was significantly decreased by CSSH treatment. IL-1β is regulated by NLRP3 inflammasome, which consists of NLRP3, caspase-1 and ASC. This complex activates IL-1β through cleavage of pro-caspase-1 and pro-IL-1β [57,58]. In HaCaT cells, we observed that CSSH inhibited NLRP3 and caspase-1, supporting its anti-inflammatory effect (Figure 5).

In acute AD phages, a Th2 dominant response occurs. However, the Th1 response is predominant in chronic AD phages; Th2 cytokines are well-known biomarkers of AD [17,59]. IL-4 and IL-13 are representative Th2 cytokines that activate the JAK1/STAT6 pathway after they bind to their intracellular receptors, IL-4R and IL-13R [60,61]. Therefore, new drugs targeting the JAK/STAT pathway were recently developed for patients with severe AD. Baricitinib, upadacitinib and abrocitinib are JAK inhibitors, whereas dupilumab, tralokinumab and lebrikizumab are IL-4/IL-13 blockers [62]. Our results demonstrate that CSSH suppresses JAK1/STAT6 activation via the inhibition of phosphorylation, suggesting that CSSH can be used to treat AD (Figure 6).

Filaggrin and involucrin, which are the skin moisturizing factors, constitute the skin epidermis, and disorders of these factors lead to water loss [24]. According to previous studies, abnormalities in filaggrin increase transepidermal water loss and contribute to AD symptoms in in vivo models. Skin barrier dysfunction, including filaggrin and involucrin deficit, collapsed tight junction and corneocyte shedding are common characteristics in AD patients [22,63]. To confirm whether CSSH protects the skin barrier, we assessed the levels of skin moisturizing factors, and found that CSSH recovered filaggrin and involucrin in TNF-α/IFN-γ-induced HaCaT cells (Figure 7). Therefore, CSSH mitigates skin inflammation via the regulation of inflammatory signaling pathways (MAPKs, NF-κB and NLRP3 inflammasome) and suppresses Th2-driven inflammatory disorder by regulating the JAK1/STAT6 pathway, as well as protects the skin barrier by restoring filaggrin and involucrin. Based on these results, we suggest that Cheungsam seed husk ethanol extract alleviates skin inflammation through the regulation of pro-inflammatory mechanisms and can be used for atopic dermatitis treatment.

## 4. Materials and Methods

### 4.1. Chemicals and Reagents

3-(4,5-Dimethylthiazol-2-yl)-2,5-Diphenyltetrazolium Bromide (MTT) was purchased from Duchefa Biochemie (Haarlem, The Netherlands). Tumor necrosis factor-α (TNF-α), and Interferon-γ (IFN-γ) were purchased from Prospec-Tany TechnoGene Ltd. (Rehovot, Israel). In addition, NLRP3, phospho-ERK1/2, ERK1/2, phospho-JNK, JNK, phospho-p38, p38, phospho-p65, p65, phospho-IκBα, IκBα, α-tubulin, Lamin B1, phospho-JAK1, JAK1, phospho-STAT6, STAT6 and β-actin were purchased from Cell Signaling Technology, Inc. (Danvers, MA, USA). Filaggrin, involucrin, caspase-1 and anti-rabbit IgG were purchased from Santa Cruz Biotechnology, Inc. (Santa Cruz, CA, USA). Goat anti-mouse IgG was purchased from AbFrontier Co. Ltd. (Seoul, Republic of Korea). For GC-MS analysis, all solvents were analytical grade ethyl acetate (EA) and trimethylsilyl (TMS) derivatization reagent N,O-bis(trimethylsilyl)-trifluoroacetamide (BSTFA) with 1% trimethylchlorosilane (TMCS) (purity ≥ 99%) were obtained from J. T. Baker (Phillipsburg, NJ, USA) and Sigma-Aldrich (St. Louis, MO, USA). Authentic standard mixture of 8 neutral cannabinoids [Cannabichromene (CBC), cannabidiol (CBD), cannabidivarin (CBDV), cannabigerol (CBG), cannabinol (CBN), delta8-THC, delta9-THC, and tetrahydrocannabivarin (THCV)] and mixture of 6 acidic cannabinoids [Cannabichromenic acid (CBCA), cannabidiolic acid (CBDA), cannabidivarinic acid (CBDVA), cannabigerolic acid (CBGA), Δ9-Tetrahydrocannabinolikc acid A (THCA-A), and tetrahydrocannabivarinic acid (THCVA)] were purchased from Cerilliant (Round Rock, TX, USA). Internal standard Δ9-THC-d9, CBD-d9 and CBDA-d9 were purchased from Cayman (Ann Arbor, MI, USA) and LGC standard (Wesel, Germany), respectively.

### 4.2. Preparation of Cheungsam Seed Husk Extract

Cheungsam seed husk was purchased from Jayhempkorea (Andong, Republic of Korea). Cheungsam seed husk 500 g was extracted with 100% ethanol 3000 mL on a mantle at 40 °C for 1 h using reflux method and repeated twice. Subsequently, the solvent was filtered using qualitative filter papers. The used filter papers were qualitative filter paper No. 1 (pore size 6 μm) and was purchased from ADVANTEC (Tokyo, Japan). After filtration, the extract was evaporated under reduced pressure at 40 °C using a rotary evaporator to remove the solvent, and then the extract was freeze-dried. Finally, 341.48 g of dried extract powder was obtained and stored at −20 °C until use. For cell treatment, 100 mg of the powder was dissolved with 1 mL of DMSO and then diluted via serial dilution method with cell culture medium, and the final content of DMSO was less than 0.1% (*v*/*v*).

### 4.3. Gas Chromatography-Mass Spectrometry Analysis of Cheungsam Seed Husk Extract

Cheungsam extract (50 μL) was transferred into a 1.5 mL vial and evaporated until dry under nitrogen (N_2_). For TMS derivatization, EA (30 µL) was added to the dried extract sample, vortex-mixed for 30 sec, and then BSTFA/TMCS (20 µL) was added; the resulting solution was heated at 80 °C for 20 min. A total of 1 µL of derivatized solution was injected into gas chromatography-mass spectrometer (GC-MS). To detect cannabinoids in the *C. sativa*, the extract was analyzed using GC-MS. GC-MS analysis was performed using an Agilent 6890N gas chromatography equipped with a DB-5MS column (30 m × 0.25 mm i.d., 0.25 μm film thickness; J&W Scientific, Folsom, CA, USA) connected to an Agilent 5975 mass spectrometer (EI mode, 70 eV). The sample was injected into the injection port and heated at 280 °C in split mode (10:1). As a carrier gas, helium (purity: 99.999%) was set at a flow rate of 1 mL/min. The oven temperature program was controlled as follows: 120 °C (0.5 min) to 200 °C (1 min) at 20 °C/min, ramped to 280 °C at 5 °C/min, and increased to 300 °C (5.5 min) at 10 °C/min. The mass spectrometer was operated in scan mode (m/z 50–600) with electron ionization energy at 70 eV. Ion source and transfer line were set at 230 °C and 250 °C, respectively.

### 4.4. Cell Culture

The HaCaT (human keratinocyte) cell line was purchased from cell lines service GmbH (Eppelheim, Germany). The cells were maintained in Dulbecco’s Modified Eagle’s Medium containing 10% fetal bovine serum, 10 mM HEPES, 1% of penicillin (1 × 10^4^ units/mL), and streptomycin (1 × 10^4^ µg/mL), and were cultured in 5% CO_2_ atmosphere at 37 °C.

### 4.5. Cell Viability Assay

Cytotoxicity of Cheungsam seed husk ethanol extract (CSSH) was assessed using MTT assay. HaCaT cells were seeded in a 96-well plate at a density of 1 × 10^4^ cells/well and were incubated for overnight in 5% CO_2_ atmosphere at 37 °C. The cells were exposed to 1, 2, 5, 10, 20, 50, 100, 200 and 500 µg/mL of CSSH for 24 h. Subsequently, cell supernatant was removed and 100 µL of MTT at 50 µg/mL was added to each well, and the cells were incubated for 4 h. DMSO was added in the plate to dissolve the crystal, and the absorbance was measured at a 540 nm of wavelength using a SpectraMAX 190 microplate reader (Molecular Devices, San Jose, CA, USA).

### 4.6. RNA Extraction and Real-Time Quantitative PCR

Total RNA was isolated from HaCaT cells using a RiboEX reagent (GeneAll Biotechnology Co., Ltd., Seoul, Republic of Korea). The cDNA was synthesized from 2 µg of RNA according to the manufacturer’s protocol using the HelixCript Easy cDNA Synthesis kit (NanoHelix Co., Ltd., Daejeon, Republic of Korea). After that, synthesized cDNA was amplified using RealHelix Premier qPCR kit (NanoHelix Co., Ltd., Daejeon, Republic of Korea), and this process was carried out using StepOnePlus real-time PCR System (Applied Biosystems, Foster City, CA, USA). The primer sequences used for PCR were listed in Table 2, and the levels of mRNA were normalized to GAPDH.

### 4.7. Western Blotting

The whole proteins were harvested and extracted using protein extraction solution (RIPA) containing 50 mM Tris-HCl (pH 7.5), 150 mM NaCl, 1% NP-40, 0.5% deoxycholic acid, 0.1% sodium dodecyl sulfate (SDS) and 1 mM PMSF. Then, SDS-PAGE was performed for protein separation, the proteins were transferred to polyvinylidene fluoride (PVDF) membranes. For blocking, the membranes were soaked in 5% skim milk dissolved in phosphate-buffered saline with 0.1% Tween 20 (PBST) at room temperature for 1 h. Subsequently, the membranes were incubated with diluted primary antibodies (1:1000) overnight at 4 °C. All antibodies used in the experiments were diluted according to the supplier’s recommendations. The membranes were rinsed three times in PBST and incubated with the corresponding horseradish peroxidase (HRP)-conjugated secondary antibodies (1:2000) at room temperature for 1 h. After washing three times in PBST, the membranes were reacted with enhanced chemiluminescence (ECL) reagents and detected using the ChemiDoc imaging system (Bio-Rad Laboratories, Inc., Hercules, CA, USA).

### 4.8. Statistical Analysis

All data are shown as mean ± standard deviation (SD). The comparisons were calculated using one-way analysis of variance (ANOVA) and followed by Tukey’s multiple comparison test to compare the differences between the control group and various groups. All the experiments were repeated three times, and all analyses were performed using GraphPad Prism version 8.0 (GraphPad software, San Diego, CA, USA). The values of *p* < 0.05 were considered statistically significant and *p*-values were given as follows: * *p* < 0.05, ** *p* < 0.01, *** *p* < 0.001.

## 5. Conclusions

This study demonstrates the potential effects of Cheungsam on atopic dermatitis in human keratinocytes. Cheungsam seed husk extract inhibited the expression of pro-inflammatory and AD-related cytokines/chemokines. In addition, Cheungsam seed husk extract reduced the phosphorylation of MAPKs and JAK1/STAT6, decreased NF-κB translocation and NLRP3 inflammasome activation, and increased filaggrin and involucrin in TNF-α/IFN-γ-induced HaCaT cells. The extract exerted anti-inflammatory and anti-atopic effects through mechanism regulation and skin barrier recovery. Therefore, Cheungsam seed husk extract may be useful for treating atopic dermatitis as well as other skin inflammatory diseases.

## Figures and Tables

**Figure 1 plants-13-01704-f001:**
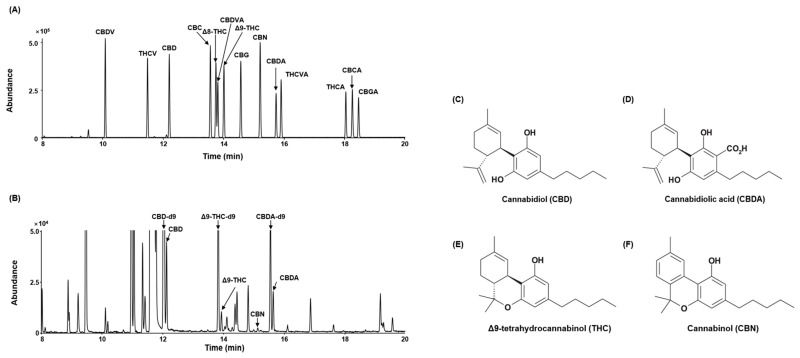
GC-MS analysis of CSSH. The TIC (Total Ion Chromatogram) of (**A**) cannabinoid standard and (**B**) CSSH. The CBD-d9, CBDA-d9 and Δ9-THC-d9 are an isotopically labeled internal standard. Cannabinoid structure of (**C**) CBD, (**D**) CBDA, (**E**) Δ9-THC and (**F**) CBN.

**Figure 2 plants-13-01704-f002:**
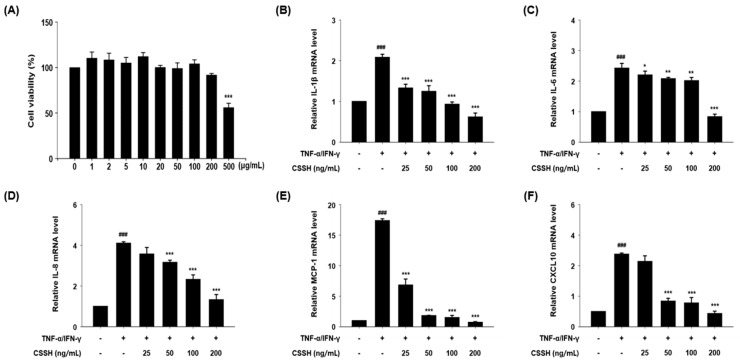
The mRNA expression of pro-inflammatory cytokines and chemokines in HaCaT cell after CSSH treatment. (**A**) Cell viability was measured using MTT assay. The 0 μg/mL is the solvent control (0.1% DMSO with cell culture medium). HaCaT cells were pre-treated with CSSH extract (25, 50, 100 and 200 ng/mL) for 30 min and then stimulated with TNF-α/IFN-γ (each 10 ng/mL) for 2 h. The mRNA levels of (**B**) IL-1β, (**C**) IL-6, (**D**) IL-8, (**E**) MCP-1 and (**F**) CXCL10 were measured using Quantitative real-time PCR and normalized to GAPDH. The data were analyzed using one-way ANOVA followed by Tukey’s post hoc comparison test between different groups. Data represent as mean ± SD of the three independent experiments. ^###^ *p* < 0.001 vs. control group; * *p* < 0.05, ** *p* < 0.01 and *** *p* < 0.001 vs. TNF-α/IFN-γ stimulated group.

**Figure 3 plants-13-01704-f003:**
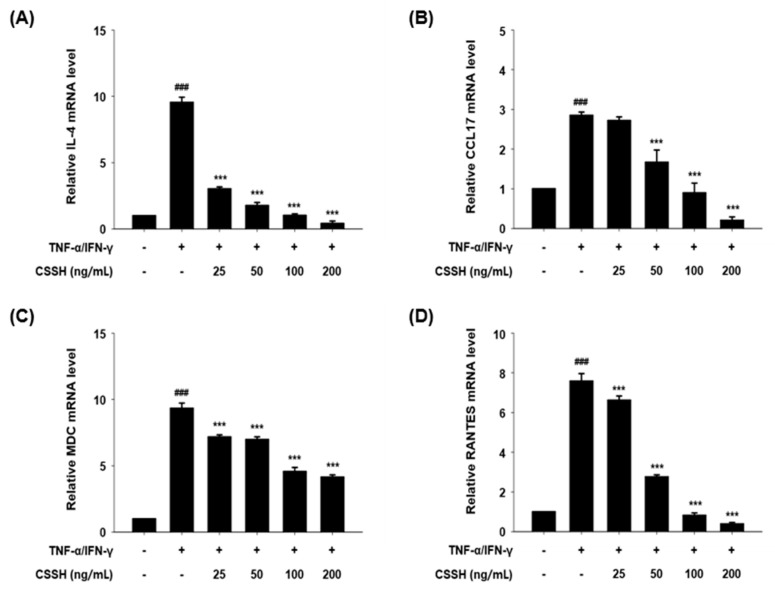
The mRNA expression of atopic dermatitis-related cytokines and chemokines in HaCaT cell after CSSH treatment. HaCaT cells were pre-treated with CSSH extract (25, 50, 100 and 200 ng/mL) for 30 min and then stimulated with TNF-α/IFN-γ (each 10 ng/mL) for 2 h. The mRNA levels of (**A**) IL-4, (**B**) CCL17, (**C**) MDC and (**D**) RANTES were measured using Quantitative real-time PCR and normalized to GAPDH. The data were analyzed using one-way ANOVA followed by Tukey’s post hoc comparison test between different groups. Data represent as mean ± SD of the three independent experiments. ^###^ *p* < 0.001 vs. control group; *** *p* < 0.001 vs. TNF-α/IFN-γ stimulated group.

**Figure 4 plants-13-01704-f004:**
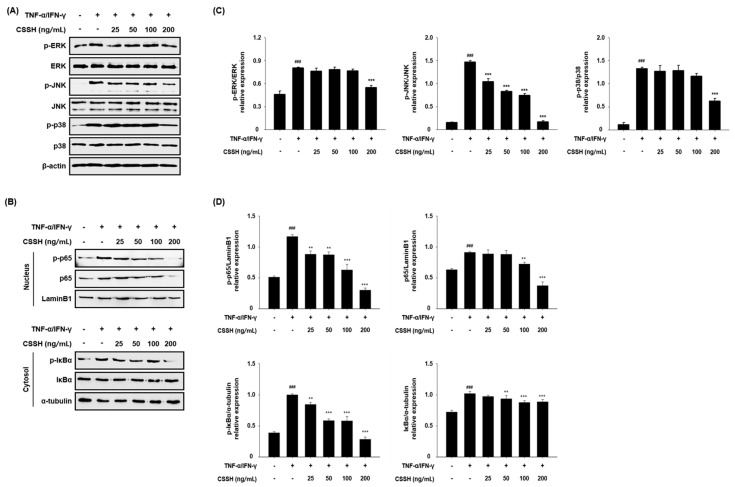
Inflammation inhibitory effect of CSSH through MAPKs inactivation and NF-κB translocation control. HaCaT cells were pre-treated with CSSH extract according to concentration for 30 min and then stimulated with TNF-α/IFN-γ (each 10 ng/mL). The protein levels of (**A**) MAPK (ERK, JNK, and p38), (**B**) nuclear and cytoplasmic p65 and IκBα were measured using Western blot analysis. The graphs are as follows: the relative expression of phosphorylated forms for total forms (**C**), the relative expression of p65 and IκBα to LaminB1 or α-tubulin (**D**). The data were analyzed using one-way ANOVA followed by Tukey’s post hoc comparison test between different groups. Data represent as mean ± SD of the three independent experiments. ^###^ *p* < 0.001 vs. control group; ** *p* < 0.01 and *** *p* < 0.001 vs. TNF-α/IFN-γ stimulated group.

**Figure 5 plants-13-01704-f005:**
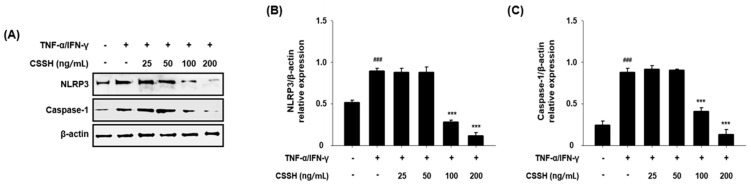
NLRP3 Inflammasome inhibitory effect of CSSH. HaCaT cells were pre-treated with CSSH extract (25, 50, 100 and 200 ng/mL) and then stimulated with TNF-α/IFN-γ for 24 h. (**A**) The protein levels of NLRP3 and caspase-1 were assessed using Western blot analysis. The graphs for relative expression to β-actin are as follows: (**B**) NLRP3, (**C**) caspase-1. The data were analyzed using one-way ANOVA followed by Tukey’s post hoc comparison test between different groups. Data represent as mean ± SD of the three independent experiments. ^###^ *p* < 0.001 vs. control group; *** *p* < 0.001 vs. TNF-α/IFN-γ stimulated group.

**Figure 6 plants-13-01704-f006:**
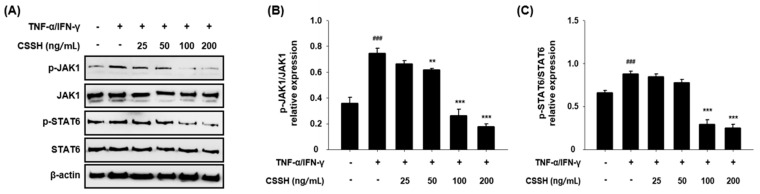
Th2 cell suppressive effect of CSSH through JAK1/STAT6 pathway control. HaCaT cells were pre-treated with different concentrations of CSSH extract (25, 50, 100 and 200 ng/mL) and then stimulated with TNF-α/IFN-γ. (**A**) The protein levels of total and phosphorylated forms of JAK1/STAT6 were measured using Western blot analysis. (**B**,**C**) The band intensity of JAK1/STAT6. The data were analyzed using one-way ANOVA followed by Tukey’s post hoc comparison test between different groups. Data represent as mean ± SD of the three independent experiments. ^###^ *p* < 0.001 vs. control group; ** *p* < 0.01 and *** *p* < 0.001 vs. TNF-α/IFN-γ stimulated group.

**Figure 7 plants-13-01704-f007:**
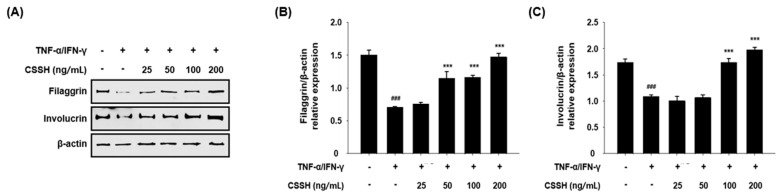
Skin barrier upregulation of CSSH via filaggrin and involucrin recovery. HaCaT cells were pre-treated with different concentrations of CSSH extract (25, 50, 100 and 200 ng/mL) and then stimulated with TNF-α/IFN-γ for 24 h. (**A**) The protein levels of filaggrin and involucrin were measured using Western blot analysis. (**B**,**C**) The band intensity of filaggrin and involucrin. The data were analyzed using one-way ANOVA followed by Tukey’s post hoc comparison test between different groups. Data represent as mean ± SD of the three independent experiments. ^###^ *p* < 0.001 vs. control group; *** *p* < 0.001 vs. TNF-α/IFN-γ stimulated group.

**Table 1 plants-13-01704-t001:** The cannabinoid content of CSSH.

Cannabinoid	CBD	CBDA	Δ9-THC	CBN
Content (μg/mg)	4.30 ± 0.065	4.25 ± 0.051	3.18 ± 0.104	3.69 ± 0.083

**Table 2 plants-13-01704-t002:** Primer sequences for real-time quantitative PCR.

Gene	Forward Primer	Reverse Primer	Amplicon Size (bp)
*IL-1β*	CTCTCTCACCTCTCCTACTCAC	ACACTGCCTACTTCTTGCCCC	93
*IL-4*	ACATTGTCACTGCAAATCGACACC	TGTCTGTTACGGTCAACTCGGTGC	113
*IL-6*	CTCCACAAGCGCCTTCGGTC	TGTGTGGGGCGGCTACATCT	121
*IL-8*	ACCGGAGCACTCCATAAGGCA	AGGCTGCCAAGAGAGCCACG	87
*MCP-1*	TCTGTGCCTGCTGCTCATAG	CAGATCTCCTTGGCCACAAT	72
*CXCL10*	TTGCTGCCTTATCTTTCTGACTC	ATGGCCTTCGATTCTGGATT	148
*CCL17*	CCATTCCCCTTAGAAAGCTG	CTCTCAAGGCTTTGCAGGTA	124
*MDC*	TGCCGTGATTACGTCCGTTAC	AAGGCCACGGTCATCAGAGTAG	129
*RANTES*	CGCTGTCATCCTCATTGCTA	GCACTTGCCACTGGTGTAGA	143
*GAPDH*	GAAGGTGAAGGTCGGAGT	GAAGATGGTGATGGGATTTC	133

## Data Availability

The original contributions presented in the study are included in the article, further inquiries can be directed to the corresponding author.

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
