# Peer review of "Cheungsam Seed Husk Extract Reduces Skin Inflammation through Regulation of Inflammatory Mediator in TNF-α/IFN-γ-Induced HaCaT Cells"

_plants, 2024, doi:10.3390/plants13121704_

Round 1

Reviewer 1 Report

Comments and Suggestions for Authors

The manuscript was titled: Cheungsam seed husk extract reduces skin inflammation through regulation of inflammatory mediator in TNF-α/IFN-γ-induced HaCaT cells.  Authors   investigated Cheungsam seed husk ethanol extract whether it has an impact on skin inflammation. The research methodology was based on the effect of the extract on changing the expression of the genes and proteins responsible for  expression levels of inflammatory mediators.

The work is interesting, but requires some corrections:

The abstract lacks information whether the research is being conducted for the first time on Cheungsam seed husk or whether there were any grounds for this type of research? What is new in the research presented by the authors?

Line 11- why the Stem  is  capitalized ?

Please correct the Latin names in italics throughout the manuscript and add a dot after the L.

Introduction: Please verify the definition of natural products of Cannabis sativa Linn.    https://doi.org/10.3390/molecules27051689 

Lines 27-28- The sentence is illegible, please verify

lines 37-39- The sentence is illegible, please verify, what is the meaning of "a few parts"- of what? What is the connection with stem?

there is no citations confirming research on other parts of Cannabis sativa Linn. for skin lesions.

Material and methods: line 99- explain where was final concentration of DMSO

What amout of seeds was used for extraction procedure?

Results-  There is no Table 2 in the manuscript - line 170

Table 1 - SD is missing, ug/mg of what? dry powder?

Where was the extract kept before experiments, at what temperature

Please: ml change to mL in whole manuscript

line 130- threre is only ")" where is "(" ?

Statistical analysis - how many repits were done?

lines 283-286- Was the amout of compounds high in comparison to other extracts obtained from leaves or flowers of Cheungsam? Change number of the Table.

Line 306 Figure- capital letter

In the Discussion section there is no reference of the results to other tested plant raw materials, including Canabis. Are the results obtained good, bad or promising?

Conclusions - What is new/ done for the first time

A lot of work was done, but overal significance is missing 

Author Response

We have responded to the reviewer's comments.

Reviewer 2 Report

Comments and Suggestions for Authors

1.       Syntax and style errors must be improved (e.g. Line 24: These natural products have been reported to have…) The manuscript needs to be brushed up!

2.       The extract obtainment lacks on basic information. Please explain how much material was used for extraction and yields of the alcoholic extract. What method was used for removing solvent at low pressure?

3.       What method assures authors that their extracts will have similar metabolite content. Please explain carefully in this section.  

4.       Please give more details on how the stock solution were prepared from lyophilized material using 0.1% DMSO a vehicle.

5.       Please add the standard used to quantify the metabolite abundance in GC-MS assays. If authors do not count with this information, please state what method was used to normalize chromatograms and quantify compounds. Several doubts earns from the poor information provided by authors.

6.       In cell viability assay there is not enough information on what controls of viability and non-viability were used. There is no information on how the percentage of viability was calculated and the role of DMSO specified in lines 125-126. A detailed method should be stated.

7.       Legend of Figures are ambiguous and grammatically incorrect (e.g. legend of Figure 1). Please check all legends accordingly.

8.       CCSH was not fully described at the beginning.  This is terrible since it causes a significant delay for readers to understand the context. Please check all abbreviations are fully described before use.

9.       The conditions for qPCR are completely missing in section 2.6 and the expected amplicon size. Authors should include a Table to organize their information.

10.   Western blotting are really poor, there is not any information on what amount of antibodies were used to perform the reaction and who provided those antibodies.

11.   The effect of several standardized extracts and individual natural products from C. sativa have shown weak or moderate cytotoxic activity on diverse cancer cell lines. Finding from authors are not new and provide new information for readers. Beyond the possible use of C. sativa seed husks as a potential source of bioactive molecules all the investigation lacks in originality.

12.   No pure compounds were tested in the systems. Why if authors count with some standards?

13.   The main weakness of this work is the fact that authors are apparently not working with standardized extracts and pure compounds to endorse their hypothesis. In addition, the husks yields from C. sativa seeds is pretty low being not a sustainable alternative to be considered. It is actually more expensive obtaining this kind of extracts than those from intact aerial parts.  Authors should explain in detail why their work provides novel information to readers and what are the possible advantages of using waste as active material.

14.   I suggest rejection with the possibility to address suggested observations.

Author Response

(The authors gave the same response as above.)

Reviewer 3 Report

Comments and Suggestions for Authors

Line 13, 'As a result' is not appropriate to start explaining the results of the current study. There are some instances where the authors have used this term which is not relevant.

Introduction:

1. No mention of any traditional use of the seeds or husk for skin conditions. 

2. There is lack of novelty of the work as the hypothesis is not backed up by the traditional use. No reference is evident on similar work or similar use.

3. It is not very clear why the husk has been selected, and it is also not very clear that how the seeds are used whether the seeds are dehulled for use and the husk comes as by product.

It needs to be clearly mentioned.

Line 30, CBD has been reported to have benefits........ what type of benefit??

Line 40: Sentence is difficult to follow.

Line 42-43: the sentence is not clear.

Lines, 50, 59,60,64-66 need to be revised, very difficult to follow and not clear what the authors are trying to explain.

Materials and Methods:

Line 71 mentions that Antibiotic-Antimycotic (100X) was used, again in Line 118 it is mentioned that Streptomycin and penicillin are used, which one is correct? 

Line 96, why only 100% ethanol is used, is there any particular reason for using only one solvent, why using two or different solvents was not considered?

Line 105: Detect cannabinoids in the C. Sativa.........? what does this mean?

Line 113, what mode positive or negative?

Line 120-127, what reference has been used for selecting the cell number and also what reference was used for the assay?

Line 123, several concentrations?? the concentrations need to be clearly mentioned here and also what solvent was used to prepare the dilutions? 

What standard has been used to compare the data of the husk extract is also not very clear.

Line 130, 131, needs revision, there is unnecessary brackets, and grammatical error.

Line 149: what is RIPA?

How many replicates used for this study is not clear?

Line 162: .......differences between these groups....... what groups?

Line 163: The version of the GraphPad used is age old, the analysis needs to be re-checked using the latest version.

Results:

To be consistent, please keep using the passive voice throughout the manuscript, (Line 168,169).

Line 177: It's not 'To confirmed'...., it should be 'To confirm...'

Again in Line 177, the concentrations need to be mentioned in full...

There are some mentions of "*p < 0.05, **p < 0.01, ***p 270 < 0.001 vs. TNF-α/IFN-γ (10 ng/ml) and ###p< 0.001 vs. negative control." throughout the manuscript, this is not clear, it needs to be explained a bit more for the readers to understand.

Line 199: ....then were induced inflammation throught...... what does this mean? please correct.

'Inhibits' is miss-spelled throughout the manuscript as 'Inhibites'

Line 232: 'To researched'...... it should be 'To research'...

In appropriate use of as a result in Line 234..

Line 248, check grammar

Discussion:

Discussion section needs to be improved as there is no mention of previous literature and comparison.

Line 275: Remove "the" after AD,...

What are the side effects that are of concern, would be better to elaborate and also the rationale for using plant extract which is better in terms of reducing those particular side effects.

Line 294: RANTES/CCL5 plays an important in allergic... what is this important? role? or something else, please revise the sentence.

Line 304: CSSH would reduce, not reduced.

Line 306: As shown in Figure 2, not As show in figure 2.. please revise

Line 315: after binding receptor in cell... what does this mean? please revise

Line 322: disorder of these lead to water loss.... not clear, please revise./

Line 324: symptom in vivo models,........ symptoms in in vivo models?!

Conclusion: The work is conducted in in vitro model and the conclusion stated by the authors is ambitious. Again, the lines 342-345 are not clear. 

Comments on the Quality of English Language

The English of the manuscript needs to be addressed seriously. There are numerous sentences which have grammatical errors and need correction. There are spelling mistakes and inappropriate use of phrasal verbs throughout the manuscript. 

Author Response

(The authors gave the same response as above.)

Round 2

Reviewer 2 Report

Comments and Suggestions for Authors

Dear authors the following points were not properly addressed nd the paper shoud be rejected. 

Point 4. According to authors, 100 mg of powder were dissolved in DMSO. This is not a corect procedure since DMS should be used after solvent extraction and evaporation. DMSO is used to resuspend hydrophobic metabolites in order to faciitate their miscibility in water. 100 mg of powder have negligible amounts of bioactive material. Then, here is a big inconsistencie that makes this ivestigation questionable.

Point 5. Authors does not explain who they normalize chromatograms by using an internal or external standard. Calibration curves with authenthic standards are the second step after adjusting the peaks to a internal or external standard. I note that authors do not know how to do this procedure and their metabolite calculations are still wrong.

Point 6. DMSO is actually toxic for the cell line assayed as well as for many other cell types. Curves with sole DMSO are required to demonstrate that the effect claimed by authors.

Point 13. I am afraid that authors are providing  wrong arguments. They are are failling in providing solid data in stock preparations and biological assays using non-standardized extracts. The use of Cannabis and their derivatives is not only restricted in Korea, most countries have certain restriction degree rules for Canabis including USA. Then authors explanation is not enough to justify their work. 

Comments on the Quality of English Language

The paper is still having syntax and grammar errors

Author Response

Thank you for your comment.

Reviewer 3 Report

Comments and Suggestions for Authors

The authors have done a great job in revising the manuscript. However, there are some spelling mistakes still in the manuscript. I would recommend the authors to revise the manuscript for spelling mistakes.

Comments on the Quality of English Language

The English language quality has improved in the revised manuscript.

Author Response

We reviewed and revised our manuscript.

We appreciate your comments on our manuscript.